# Optimization and Characterization of Protein Extraction from Asparagus Leafy By-Products

**DOI:** 10.3390/foods13060894

**Published:** 2024-03-15

**Authors:** Aline Cristini dos Santos-Silva, Bianka Rocha Saraiva, Anderson Lazzari, Henrique dos Santos, Évelin Lemos de Oliveira, Francielle Sato, Eduardo César Meurer, Paula Toshimi Matumoto-Pintro

**Affiliations:** 1Food Science Graduate Program, State University of Maringá, Maringá 87020-900, Brazil; alinecristinisantossilva@gmail.com (A.C.d.S.-S.); andersonlazzari29@gmail.com (A.L.); 2Animal Science Graduate Program, State University of Maringá, Maringá 87020-900, Brazil; bianka_saraiva@hotmail.com; 3Physics Graduate Program, State University of Maringá, Maringá 87020-900, Brazil; rique.lovo@gmail.com (H.d.S.); fsato@uem.br (F.S.); 4Chemistry Department, Federal University of Paraná, Jandaia do Sul 86900-000, Brazil; evelynlemos@hotmail.com (É.L.d.O.); eduardo.meurer@gmail.com (E.C.M.)

**Keywords:** protein isolate, extraction process, vegetable proteins

## Abstract

Asparagus production generates significant amounts of by-products during the summer and post-harvest growth period. By-products can be good sources of nutrients and phytochemicals. The interest in increasing the availability of proteins for human consumption has led to the use of new plant sources rich in proteins. The objective of this study was to use response surface methodology (RSM) to optimize the aqueous extraction process of proteins from asparagus leafy by-products, for the production of new protein ingredients. The optimum extraction condition was at pH 9, with 40 min of extraction at 50 °C, and the concentration was fixed at 5 g·L^−1^. The isolate obtained presented 90.48% protein with 43.47% protein yield. Amino acids such as alanine, proline, valine, leucine/isoleucine, asparagine, and phenylalanine were identified, and the antioxidant activity for 2,2 AZINO BIS (3-ethylbenzo thiazoline 6 sulfonic acid diammonium salt) was 145.76 equivalent to Trolox μmol.100g^−1^ and for DPPH 65.21 equivalent to Trolox μmol.100g^−1^. The product presented favorable technological properties (water absorption capacity 4.49 g·g^−1^ and oil absorption capacity 3.47 g·g^−1^) and the color tended towards dark green (L* 31.91, a* −1.01, b* −2.11). The protein isolate obtained through the extraction optimization process showed high potential to be used as a protein ingredient.

## 1. Introduction

Asparagus (*Asparagus officinalis* L.), popularly known as the “king of vegetables”, is a perennial herbaceous vegetable belonging to the Asparagaceae family. Asparagus spears are considered a healthy food due to their low-calorie content, high protein, and bioactive phytochemical content. Once the asparagus shoots begin to open, they quickly become woody and are not commercially accepted. After this period, the spears can transform into ferns with modified stems, called cladophylls or cladodes, which are photosynthetically active, leaf-like organs [1].

New applications for agricultural and industrial waste have been investigated, aiming at a positive environmental impact and transformation into usable compounds. The plant commercial part corresponds to the asparagus edible stem, which comprises less than 25% of the plant; the remaining 75% includes the hard stem, root, and leaves. These fractions are characterized as by-products, but the same nutrients and phytochemicals present in the spear can be found in them, and, therefore, these by-products have high potential for use as food ingredients with nutritional value and health-promoting properties [2].

According to FAO estimates, by 2050 [3], the world population will reach around 9.6 billion, and it is estimated that global food production will increase by 50%. It is necessary to obtain new food alternatives to ensure food security; therefore, new sources of vegetable proteins have been researched. Proteins of animal origin have limited availability and high cost; in addition, consumers have also associated the excessive consumption of animal-origin meat with health problems. Therefore, the use of proteins of vegetable origin, generally of a lower cost and greater availability, can become an alternative to meet this demand [4].

In developing countries, cereals and legumes are the most important dietary protein sources, and vegetables help meet the population’s protein nutritional needs [5]. The challenge is to convert these plant sources into acceptable and functional protein ingredients.

Protein isolates have gained interest due to the quality of the isolated protein and its versatility in the preparation of traditional foods or in the development of new foods [6]. The success of using plant protein isolates depends on their functional properties, which are influenced by intrinsic factors (protein composition and conformation), environmental factors (food or model system composition), and the methods and conditions used during isolation [7].

Among the methods used to produce the isolate, isoelectric precipitation is a traditional method that consists of protein extraction by diluted alkali, followed by precipitation at the isoelectric point. Parameters such as pH, temperature, time, ionic strength, and solvent/meal ratio can significantly affect protein extraction capacity [8]. Response surface methodology (RSM) can be used when many factors and interactions affect the desired response, as this tool provides relevant information in less time and with a reduced number of experiments. Optimal operating conditions are achieved using more complex experimental designs such as Box–Behnken Designer (BBD), a class of rotating or quasi-rotating second-order designs based on three-level incomplete factorial designs [9].

By-products have been studied in juice production, the processing of dietary fiber powder, and the extraction of bioactive compounds [2], but limited studies have been reported on the valorization of the leafy by-products of asparagus, used only as animal feed or ground cover for the next harvest. Protein extraction from asparagus by-products may present a viable and sustainable approach to meet the growing demand for alternative protein sources, offering a new solution to increase nutritional value, reduce waste, and contribute to the development of environmentally friendly protein supplements. The objective of the study was to use RSM to optimize the protein extraction process from asparagus leafy by-products and evaluate the physicochemical and technological properties of the protein isolate obtained.

## 2. Materials and Methods

### 2.1. Materials

Leafy by-products or cladodes that developed after harvesting asparagus were collected in Marialva, Paraná, Brazil (23°29′8″ S, 51°47′34″ W). They were sanitized and dried in an oven with air circulation at 55 °C for 18 h, and then ground and sieved using a 60 mesh. The powder obtained was stored away from light at room temperature and was called asparagus by-product.

### 2.2. Asparagus By-Product Characterization

The asparagus by-product powder was analyzed in triplicate for moisture, ash, lipid, crude protein, and crude fiber content according to the Adolfo Lutz Institute [10]. The moisture content was determined through drying in an oven at 105 °C, until constant weight, and the ash content through incineration in a muffle furnace at 550 °C for 4 h. The amount of total lipids was extracted using Soxlet and nitrogen using the Kjedahl method. Carbohydrate content was determined by subtracting the sum of moisture, protein, fat, fiber, and ash percentages from 100%.

The protein fractionation of the asparagus by-product was based on its solubility in water, saline, hydroalcoholic, and alkaline solutions [11]. The powder was mixed with deionized water (1:10 *w*/*v*), homogenized for 30 min, and centrifuged at 2249× *g* for 20 min. The supernatant obtained was called albumin fraction. The pellet was successively resuspended in different solutions under the same conditions described for the albumin fraction. The 0.90% (*w*/*v*) NaCl solution was used to solubilize the globulin fraction, the 70% (*w*/*v*) ethanol solution for prolamin, and the 0.10 M NaOH solution for glutelin. The following equations were used:(1)$$\mathrm{Total}\;\mathrm{protein}\;(\%)=\frac{\mathrm{Total}\;\mathrm{protein}\;(g)\;\mathrm{of}\;\mathrm{each}\;\mathrm{fraction}}{\mathrm{Total}\;\mathrm{proteins}\left(g\right)\;\mathrm{of}\;\mathrm{asparagus}\;\mathrm{by-product}\;\mathrm{powder}}\times 100$$
(2)$$\mathrm{Insoluble}\;\mathrm{residues}=100-\left(\mathrm{albumins}+\mathrm{globulins}+\mathrm{prolamins}+\mathrm{glutelins}\right)$$

### 2.3. Protein Extraction and Experimental Design

The asparagus by-product was extracted into deionized water using the combination of independent variables: pH (8–10), time (20–60 min), and temperature (30–70 °C). The concentration was not used as a variable, but it was fixed at 5 g·L^−1^ for all runs after preliminary tests. After pH adjustment, the samples were subjected to a water bath with agitation under predetermined conditions. They were then centrifuged (MPW-351R Med. Instruments, Boremlowska, Poland) at 5046× *g* for 20 min at 25 °C, filtered through a quantitative filter (quantitative paper, weight 80 g), and the supernatant was collected and denominated protein extract.

The protein content evaluated in the protein extract was denominated response variable (Y). It was evaluated by homogenizing the supernatant (5 µL) with Bradford reagent (250 µL), and after 10 min of reaction, the absorbance was read in ELISA microplate readers (Molecular Device Versa Max, Canton, MA, USA) at 595 nm. An analytical curve was constructed using bovine serum albumin [12].

RSM using the Box–Behnken design was used to optimize the extraction process. The effect of three independent variables (X_1_, pH; X_2_, extraction time; and X_3_, extraction temperature) at three levels was performed to determine the effect on the dependent response variable extracted protein content. The variables and their levels were defined based on preliminary unpublished tests within the range of investigation. For statistical calculation, the variables were coded according to the equation:(3)$$x_{i}=\frac{X_{i}-X_{0i}}{\Delta X}=1,2,3$$
where xi is the independent variable coded value, X_i_ is the independent variable real value, X_0_ is the independent variable real value on the center point, and ΔX is the step change value of an independent variable.

The whole design, consisting of 15 experimental points, was carried out in randomized order. Two replicates at the center point were used to allow for the estimation of a pure error sum of squares. The response function investigated, Y (mg of protein^−1^ extracted from the asparagus by-product), was fitted to a second-order polynomial model. The quadratic model for predicting the optimal point was expressed according to the equation:(4)$$Y=\beta 0+\sum_{i=1}^{k}\beta iXi+\sum_{i=1}^{k}\beta iiXi^{2}+\sum_{i=1}^{k-1}\sum_{j=2}^{k}\beta ijXiXj+\varepsilon$$
where *β*_0_, *β_i_*, *β_ii_*, and *β_i_j* are the regression coefficients for intercept, linear, quadratic, and interaction, respectively. *Xi* and *Xj* are the factors, and *k* is the number of independent variables [13].

Optimal extraction conditions were determined through desirability function, aiming to maximize the extracted protein content. The response obtained under the predicted optimal condition was validated by comparison with the response predicted by the model.

### 2.4. Characterization of the Extraction Process

Extraction efficiency was determined by the protein content of the protein extracted under ideal conditions in relation to the protein content of the asparagus by-product powder before the extraction process, expressed as a percentage.

### 2.5. Isoelectric Point Determination

The isoelectric point (IP) was determined in the protein extract obtained in optimal conditions from the experimental designer. Protein extracted (5 mL) was diluted in distilled water (40 mL), and the pH was adjusted from 2–10 using 0.5 or 3 N HCl/NaOH. The turbidity of each solution was read on a spectrophotometer at 320 nm, and the pH value that presented the greatest turbidity was considered the isoelectric point [14].

### 2.6. Asparagus By-Product Protein Isolate

The protein extract obtained under optimal conditions was adjusted to the IP previously determined. The precipitated protein was recovered after centrifugation for 20 min (10,180× *g*) at 4 °C, mixed with distilled water, centrifuged again, and then the pellet was lyophilized at −51 °C (CHRIST, Alpha 1–4, LD Plus). Protein isolate (powder) was stored for later analyses. The yield of the extracted protein was calculated using the equation:(5)$$\mathrm{Yield}\left(\%\right)=\frac{\mathrm{weight}\;\mathrm{of}\;\mathrm{isolate}\times \%\mathrm{protein}\;\mathrm{content}\;\mathrm{of}\;\mathrm{isolate}}{\mathrm{weight}\;\mathrm{of}\;\mathrm{asparagus}\;\mathrm{by-product}\;\mathrm{powder}\times \%\mathrm{protein}\;\mathrm{content}\;\mathrm{of}\;\mathrm{asparagus}\;\mathrm{by-product}\;\mathrm{powder}}\times 100$$

### 2.7. Technological Properties

#### 2.7.1. Color

Color analyses were carried out with the colorimeter (Konica Minolta CR400, Tokyo, Japan) and illuminant C based on the CIELAB system, which evaluates color by light reflectance in three dimensions: L*, luminosity on a scale from 0 (black) to 100 (white); a*, scale ranging from red (0 + a) to green (0 − a); and b*, which represents a scale from yellow (0 + b) to blue (0 − b).

#### 2.7.2. Water Absorption Capacity (WAC)

The protein isolate (100 mg) was mixed with 1000 µL of distilled water using a vortex and left to rest for 30 min. The suspension was centrifuged at 1800× *g* for 20 min at 22 °C, the supernatant was discarded, and the tube was drained at a 45° angle (10 min). Water absorption capacity was calculated by rate water volume absorbed by sample weight [15].

#### 2.7.3. Oil Absorption Capacity Analyses (OAC)

The protein isolate (100 mg) was mixed with 1000 µL of sunflower oil, left to rest for 30 min (22 °C), and centrifuged at 13,600× *g* for 10 min at 25 °C. The supernatant was discarded and drained at a 45° angle for 20 min. The volume of oil absorbed was divided by the weight of the protein sample to obtain the fat absorption capacity of the sample [16].

#### 2.7.4. Protein Solubility

Protein isolate solubility was determined according to Saraiva et al. [17]. The protein isolate (200 mg) was mixed with 20 mL of distilled water, and the pH was adjusted to 2, 4, 6, 8, 10, and 12 with 0.5 or 3 N HCl/NaOH. The mixture was stirred for 60 min at 25 °C, and centrifuged at 5046× *g* for 30 min at 4 °C. The supernatant was collected, and the soluble protein content was determined by the Bradford method [12]. The following finding was used to calculate solubility:(6)$$\mathrm{Solubility}\;(\%)=\frac{\mathrm{Protein}\;\mathrm{content}\;\mathrm{in}\;\mathrm{supernatant}}{\mathrm{Total}\;\mathrm{protein}\;\mathrm{content}\;\mathrm{of}\;\mathrm{sample}}\times 100$$

### 2.8. Total Polyphenol Content (TPC) and Antioxidant Capacity

#### 2.8.1. Preparation of Extract

The protein isolate (1:100 *w*/*v*) was homogenized with ethanol/water (70:30, *v*/*v*) for 15 min. It was centrifuged at 5046× *g* for 15 min at 4 °C, and the supernatant was recovered for subsequent analyses.

#### 2.8.2. Total Polyphenol Content (TPC)

The total polyphenol content was determined using the Folin–Ciocalteau method [18]. An aliquot (125 μL) of the extract was mixed with 125 μL of Folin–Ciocalteau reagent (50%) and 2250 μL of sodium carbonate (Na_2_CO_3_) (3.79M). After homogenization, the samples remained protected from light for 30 min. The reading was carried out on a spectrophotometer at 725 nm. Absorbance was compared with a gallic acid standard curve. The results obtained were expressed in mg of EAG (equivalent to gallic acid).

#### 2.8.3. Antioxidant Capacity Using ABTS and DPPH Assays

ABTS^+^ (2,2 AZINO BIS (3-ethylbenzo thiazoline 6 sulfonic acid diammoninum salt) and DPPH (2,2-Diphenyl-1-Picrylhydrazyl) assays were carried out according to Saraiva et al. [16]. The ABTS solution (7 mM) in 100% ethanol reacted (for 16 h at room temperature) with potassium persulfate (140 mM) to form the ABTS radical cation solution (ABTS^•1^). It was diluted again in 100% ethanol until an absorbance of 0.7 ± 0.05 at 734 nm was obtained. For analysis, 40 μL aliquots of the extracts were added to 1960 μL of diluted ABTS^•1^ solution, and the absorbance was read at 734 nm after 6 min.

Aliquots of 150 µL of extract were mixed with 2.85 mL of the DPPH radical (0.06 mM), homogenized in a vortex, and kept away from light for 30 min. Readings were taken on a spectrophotometer at 515 nm.

DPPH and ABTS free-radical scavenging results were evaluated and expressed as a Trolox equivalent antioxidant capacity (TEAC) per 100 g of dry matter of sample (μmol.100g^−1^).

### 2.9. Fourier-Transform Infrared Spectroscopy (FTIR)

FTIR analyzes were performed on a Bruker Vertex 70v spectrometer (Bruker Optik GmbH, Billerica, MA, USA) with an attenuated total reflectance (ATR) accessory. The asparagus by-product powder and its protein isolate samples were separated into aliquots (*n* = 3), and then subjected to FTIR-ATR measurement. Each spectrum is the result of an average of 128 scans, measured from 4000 to 400 cm^−1^ with 4 cm^−1^ spectral resolution. Measurements were carried out at room temperature and normalized by the normalization vector using the OPUS software version 7.2 [19].

### 2.10. Qualitative Identification of Amino Acids

The amino acids were analyzed according to Poliseli et al. [20]. For sample preparation, 0.10 g of protein isolate from the asparagus by-product was dissolved in 1.0 mL of 50 mM ammonium bicarbonate solution. The solution was vortex-mixed and refrigerated at 4 °C for 60 min. It was centrifuged at 1696× *g* for 10 min and diluted (1:10 *v*/*v*) twice in mobile phase acetonitrile/water/formic acid (70:30:0.1) (*v*/*v*/*v*). The diluted sample (5 µL) was injected directly into the mass spectrometer, with electrospray ionization and triple quadrupole analyzer (Waters, XE Premier, Wexford, Irlanda).The analysis run time was 2 min for each sample. The identification of M + H ions was carried out according to Cantú et al. [21] with neutral loss-type fragmentation.

### 2.11. Statistical Analyses

The experiment was carried out three times and the analyses were performed in triplicate. The results obtained were subjected to a two-way analysis of variance (ANOVA) using the general linear model (GLM) with SPSS (v.15.0) (IBM SPSS Statistics, SPSS Inc., Chicago, IL, USA) for Windows. Averages and standard deviation were calculated for each variable. For the differences that were statistically significant, Tukey’s test was used with a significance level of 5%.

## 3. Results and Discussion

### 3.1. Asparagus By-Product Powder Properties

Asparagus by-product powder contains approximately 15% protein, and is rich in dietary fiber and carbohydrates, in addition to having low amounts of lipids and ash (Table 1). The leaves have recently been recognized as low-cost protein sources due to their ability to readily synthesize amino acids using sources such as sunlight, carbon dioxide, and atmospheric nitrogen. The leaves of several plants were investigated as a potential source of proteins: alfalfa 15–20% [22], jackfruit (*Artocarpus heterophyllus* Lam) 19.35% [23], and tobacco (*Nicotiana tabacum* L.) 9.4–14.6% [24] (protein contents dry weight basis).

The protein fractions of asparagus by-product powder are presented in Table 1. Among the sample soluble part, the glutelin fraction presented the highest content (*p* < 0.05), followed by the prolamin and albumin fractions, which presented the same proportion, and globulin with the lowest value. The content of insoluble residues shows that almost 34% of the sample is not soluble in any of the solvents used at room temperature. In protein fractions extracted from radish leaves (*Raphanussativus* L.), glutelin was also higher than the other fractions (glutelin 42.58%, prolamin 23.57%, albumin 20.08%, and globulin 13.74%) [25].

Glutelins are proteins that are poorly soluble in water but are easily solubilized in alkaline (pH > 10) and acidic (pH < 4) conditions [26]. The alkaline solution is generally efficient to solubilize proteins in both cases, as most vegetable proteins have greater solubility in the alkaline pH range, due to the higher content of amino acids such as aspartic acid and glutamic acid composed of proteins of plant origin [27]. Soluble proteins directly affect functionality during food processing. Knowing the composition of soluble proteins in asparagus by-products is extremely important, as it is decisive in choosing the extraction method that aims to extract the most predominant protein content.

### 3.2. Influence of Extraction Conditions on Soluble Protein Content

In the experimental design, the soluble protein content varied between 110.57 and 133.01 mg·g^−1^. Figure 1E,F show an increase in the amount of protein extracted with an increase in time from 20 to 40 min and temperature from 30 to 50 °C at pH 9 (116.55 to 130.99 mg·g^−1^). Changing the solution pH from 8 to 10 resulted in 9.90% more protein soluble in the extract, and the minimum protein extraction values were obtained at pH 8, 20 min, and 50 °C (Table 2).

Increasing temperature and pH (Figure 1A,B), there is an increase in protein content. This increase can be observed up to around 55 °C, above which it begins to reduce. The reduction in protein content at elevated temperatures may be associated with heat-induced protein denaturation or heat-induced gelation at temperatures above 60 °C. Above pH 10, the protein content reduces due to the unfolding or denaturation of protein molecules in a highly alkaline medium, leading to a distorted conformation with modified hydrophobicity of the protein under study. Regarding time, 40 min was adequate for extracting proteins from asparagus by-products (Figure 1C–F). The extraction time must be adequate to disperse the protein in the solvent, as a long extraction time (greater than 2 h) can result in the saturation of the protein, resulting in a reduction in protein content [28].

### 3.3. RSM and Optimization of the Extraction Condition

The regression equation obtained for protein content (Y) was as follows:(7)$$Y=-889.74+209.61X_{1}-11.53X_{3}-76X_{1}+0.64X_{1}+4.4X_{2}^{2}+0.57X_{1}-0.03X_{3}-0.08+371.8$$

According to Table 3, F values with a *p* value less than 0.05 describe that the model terms are significant when quantifying the exact response trend. The adequacy of the model, through the R-squared (0.9939) and Adj R-squared (0.9737) values, suggests that the total variation of 97.37% in protein content was attributed to the extraction conditions. The difference between R-squared and Adj R-squared is insignificant, reflecting that the polynomial model has practically no insignificant terms, indicating a high degree of precision and reliability between the experimental values.

The significance of each term through *p* values on protein content was analyzed using ANOVA. The results showed that there were differences in the model, and two linear terms (X1 and X3) and all quadratic terms were significant (*p* < 0.01), while the other terms showed no difference (*p* > 0.05). For each term presented in the models, the lower the *p* value and the higher the F value, the greater the impact of the response variable.

The three-dimensional RSM graphs (Figure 1A–F) were created with the response variable (Y) on the Z axis against any two independent variables, keeping the other variable at the central point value (coded value = 0). Through these graphs, it is possible to observe the interaction of the analyzed variables.

The optimization aimed to obtain the maximum amount of soluble protein. According to Derringer’s desirability function, a multi-criteria methodology currently widely used in the optimization of procedures [29], the optimal extraction conditions were pH 9.0, an extraction time of 40 min, and a temperature of 50 °C. The extraction was carried out in triplicate under these conditions and obtained 136.89 ± 1.49 mg·g^−1^, while the value predicted by the model was 132.96 mg·g^−1^; the values’ proximity reflects the adequacy of model validation.

The extraction process presented a yield of extracted protein of 43.47% ± 1.20, with a corresponding protein content of 90.48% ± 1.94. At pH 9.0, the protein extraction efficiency was 84.44% ± 0.67 (Table 4).

The extraction rate and total protein content depend on the by-product source, the cultivar, the processing, and the biomass type used, as well as the extraction method and the conditions applied. However, the amount of alkali and temperature are decisive factors that must be controlled to improve protein recovery [28]. Furthermore, alkaline extraction is the most widely used type of extraction due to its simplicity, speed, and low cost [30].

### 3.4. Characterization of the Isolated Protein

#### 3.4.1. Solubility, Isoelectric Point, and Qualitative Identification of Amino Acids

The extracted solution was evaluated for the ideal pH to precipitate and isolate the proteins. Proteins extracted from asparagus by-products have an isoelectric point of pH 3.2 (Figure 2). The minimum solubility of the material around pH 4, close to pH 3, in Figure 3, demonstrates electrostatic repulsion, favoring particle precipitation. The protein solubility of the isolate increased between pH 6 and 8 (Figure 3), showing pH-dependent solubility. This maximum solubility profile at alkaline pH values and minimum solubility profile at acidic pH values has also been observed in other plant proteins such as amaranth, eggplant, and pumpkin leaf protein isolates [31].

Differences found in the isoelectric points of plant proteins can be attributed to the content of hydrophilic and hydrophobic amino acids. The amino acids identified in this study (Figure 4) were mostly hydrophobic; the only hydrophilic amino acid identified was asparagine.

Figure 4 shows the mass spectra obtained in which possible combinations of di- and tripeptides with sufficient intensity to fragment were not identified (absolute intensity close to 1000), but identification was possible using m/z ions for the isolate: 90, 116, 118, 132, 133, and 166, which characterize the amino acids alanine, proline, valine, leucine/isoleucine, asparagine, and phenylalanine.

The nutritional value of proteins is related to the amino acid content, their mutual proportions, and digestibility [32]. Asparagus spears have around 20% protein (*w*/*w*, dry matter), and are abundant in most essential amino acids, comprising 40–43% of total amino acids, with glutamic acid and aspartic acid as dominant amino acids, and methionine, cystine, and leucine as limiting amino acids. About 15 amino acid types were identified in asparagus spears and presented spatial distribution [33]. The essential amino acids identified in this study were valine, leucine/isoleucine, and phenylalanine.

#### 3.4.2. Structural and Conformational Properties

Figure 5a shows the FTIR-ATR spectra of the powdered asparagus by-product samples and its protein isolate. Bands referring to the stretching of N–H (νNH) of proteins and O–H (νOH) of carbohydrates and water are observed in the region between 3000 and 3600 cm^−1^, while the peaks between 3000 and 2818 cm^−1^ can be attributed to the antisymmetric stretching of CH3 and CH2, respectively [34].

In Figure 5a, a band centered at 1731 cm^−1^ is observed only for the spectrum of powdered asparagus by-product, which may be related to the C=O (vCO) stretching of polyphenols, commonly present in asparagus [34]. The content of total polyphenols (Table 4) present in the protein isolate obtained was lower than that present in the powder (11.55 mg GAE·g^−1^). During the protein extraction process, some phenolic compounds originally present in soluble form may be lost, as well as phenolics that were part of the protein-bound fraction. Extraction carried out at pH 9 can break interactions between phenols and proteins [35].

Both spectra showed bands associated with amides I and II found between 1709 and 1476 cm^−1^, with amide I found at higher wave numbers than amide II, centered, respectively, at 1627 and 1525 cm^−1^. Furthermore, the peak centered around 1400 cm^−1^ is attributed to the asymmetric deformation of CH3 (δCH3), and finally, the region between 1300 and 900 cm^−1^ refers to C–O stretching (vCO) that is associated with carbohydrates [34].

To better evaluate the proportions of total protein (amide I) and carbohydrates in the samples, the areas of the bands referring to amide I and carbohydrates (vC-O) were obtained through integration. The comparative analyses of the areas are shown in Figure 5b. Comparing the area of the bands associated with carbohydrates, it is possible to observe a reduction in the contribution of the bands in the spectrum, while the amide I band becomes more evident after protein isolation. Therefore, the samples showed higher protein concentrations after isolation. This result can be verified in Table 1 through the chemical composition of the asparagus powder, where carbohydrates predominate over proteins.

The amide I band is highly sensitive to the molecular conformation of the protein [36]. Thus, to determine the effects caused on the structure after protein isolation, a deconvolution of the amide I band was performed using Gaussian adjustment, as shown in Figure 6a. This mathematical treatment is necessary, as the spectral contributions of the secondary structures in the amide I band are overlapped. In this way, five band centers were determined, in the amide I region, referring to the structures: β-Sheet, Random Coil, α-Helix, β -Turn centered, respectively, at 1627, 1644, 1659, 1675, and 1688 cm^−1^ [37].

From the areas obtained from the deconvolution of amide band I [Figure 6b], the asparagus by-product powder and its protein isolate present a dominance of the β-Turn structure. The results found differ from the *Moringa oleifera* protein isolate, which had a predominant secondary structure of β-Sheet (31.73%). The high proportion of β -Turn is important for protein flexibility, contributing to the stabilization of the water–oil emulsion, facilitating the globular protein emulsion [38]. The amount of Random Coil was low, suggesting that the optimal extraction conditions (Table 4) caused little tension, not promoting folding, which can be justified by the fact that the extracted protein suffered little tension during the extraction process.

#### 3.4.3. Chemical and Functional Properties

The color parameters, technofunctional properties, bioactive compounds, and antioxidant activity of asparagus by-product isolate obtained at optimum conditions are presented in Table 4.

The color of the asparagus by-product powder (59.28 ± 0.14 L*, −8.92 ± 0.11 a* and 24.06 ± 0.12 b*), demonstrated a greater tendency to luminosity (L*), it was more prone to green (a*), and tended more towards yellow (b*) when compared with the color of the isolated protein (Table 4).

The dark green color observed in the protein isolate obtained may have been influenced by pH modulation. Under alkaline conditions, the oxidation of phenolic compounds in the protein matrix can be accelerated due to the possible covalent bond of these compounds with proteins [28], since this interaction can form polymers with greater pigmentation.

The water and oil absorption indices (WAC and OAC, respectively) of the protein isolate are shown in Table 4. The results found in this study were superior to protein isolated from lupine (*Lupinus luteus* L.), alfalfa (*Medicago sativa* L.), and moringa leaves (*Moringa oleifera* L.) [39,40,41], and lower than those obtained for protein isolate from defatted chia flour [42].

Extraction conditions can influence WAC and OAC, as slightly higher temperature and pH can cause the obtained protein isolate to dissociate, providing more water binding sites in the subunits when compared to the subunits [13]. Proteins can interact with water and oil in foods due to the hydrophilic and hydrophobic properties of the polar amino acids present and their location in the molecule [43].

The total polyphenol content and antioxidant properties are shown in Table 4. The total polyphenol content for the protein isolate was higher than that found by Vital et al. (2020) [44] for non-commercial asparagus (2.33 mg GAE·g^−1^), while Garg et al. (2020) [28] found 2.8 mg GAE·g^−1^ in sangri protein concentrate. Polyphenols can be extracted together with proteins due to the ability of these two compounds to interact. This complex formation leads to the formation of soluble or insoluble compounds that can reduce or increase the antioxidant activity of polyphenols, as well as affect enzyme digestion [45].

Fruits and vegetables are the main sources of phenolic compounds in the diet, and the World Health Organization recommends that daily intake be 400 g·day^−1^ [46]. However, in everyday life, their consumption is considered low, resulting in the need to incorporate these compounds in the development of new products. Therefore, the obtained protein isolate can complement the necessary intake.

The asparagus by-product isolate showed antioxidant capacity using ABTS (145.76 ± 0.76 TEAC μmol.100g^−1^) and DPPH (65.21 ± 0.38 TEAC μmol.100g^−1^) assays. Using the ABTS method, the isolate from asparagus by-products showed greater antioxidant capacity than when using the DPPH method. This can be justified by the way the radical dissolves in the medium. Whereas ABTS is soluble in organic and aqueous media, selecting both lipophilic and hydrophilic antioxidants, DPPH dissolves only in organic media, having greater selectivity for hydrophobic systems [47]. This can also be verified through the WAC and OAC values (Table 4), where hydrophilic and lipophilic characteristics are observed in the obtained isolate.

## 4. Conclusions

The results indicate that the three variables studied (pH, time, and temperature) significantly affect the aqueous extraction process of protein from the asparagus by-product. The optimal extraction conditions were obtained when the extraction pH was 9, the time was 40 min, and the temperature was 50 °C. By optimizing the protein extraction method for asparagus by-products it was possible to obtain a protein isolate with essential amino acids; bioactive compounds with antioxidant activity present; and technofunctional properties favorable for its use as an ingredient for new products. Knowledge of the composition and structure of soluble proteins is important to elucidate their functionality and guide their future applications. In view of economic viability, more studies are needed to understand the main challenges for industries and rural producers in food processing and to value vegetable protein resources, as well as asparagus by-products.

## Figures and Tables

**Figure 1 foods-13-00894-f001:**
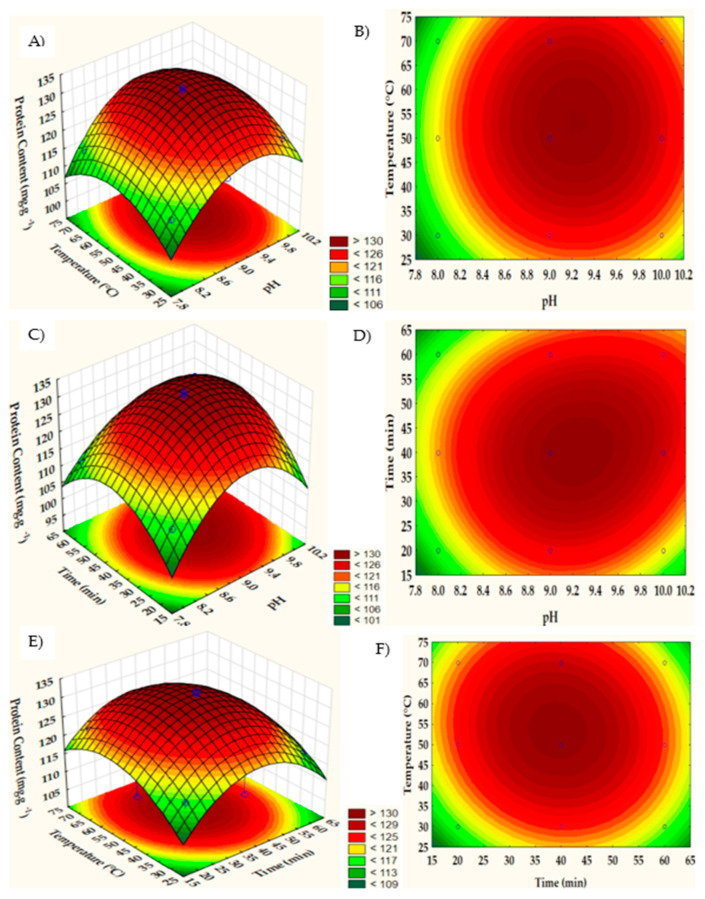
Surface and contour plots of protein extraction response from the asparagus by-product showing the effects of temperature (°C) and pH on extraction (**A**,**B**); the effects of time (min) and pH on extraction (**C**,**D**); and the effects of temperature (°C) and time (min) on extraction (**E**,**F**).

**Figure 2 foods-13-00894-f002:**
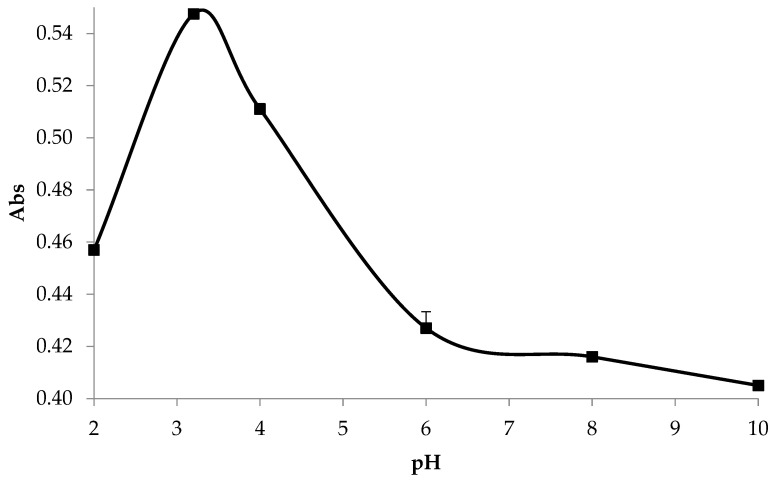
Isoelectric point of protein solution from asparagus; protein extracted determination in the pH 2–10 range.

**Figure 3 foods-13-00894-f003:**
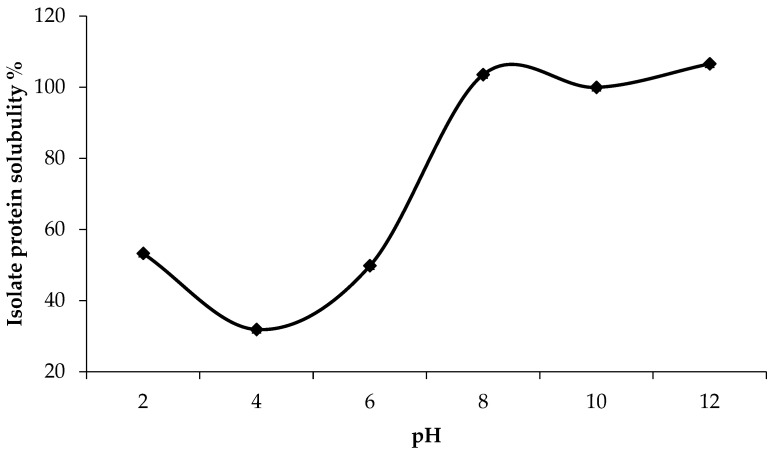
Solubility profile of protein isolate from asparagus by-product at different pH values.

**Figure 4 foods-13-00894-f004:**
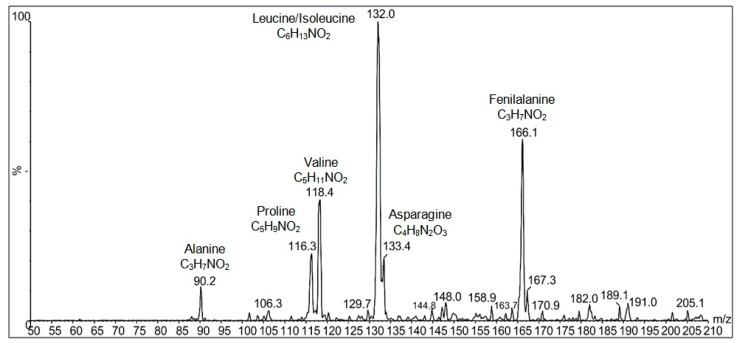
Amino acid composition of protein isolate from asparagus by-product determined by mass spectrum *m*/*z* 50 to 210 obtained by neutral loss 46 Da and identification using the DE NOVO method.

**Figure 5 foods-13-00894-f005:**
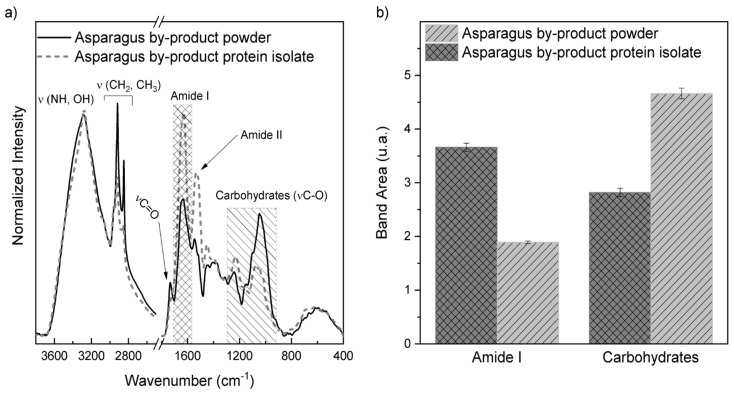
(**a**) Average FTIR-ATR spectra (*n* = 3) of powdered asparagus by-product samples and protein isolate from asparagus by-product; the hatched regions correspond to the amide I and carbohydrate bands. Highlighted in (**b**) are the areas of amide I and carbohydrate bands. Different lowercase letters are significantly different (*p* < 0.05) for amide I. Different uppercase letters are significantly different (*p* < 0.05) for carbohydrates.

**Figure 6 foods-13-00894-f006:**
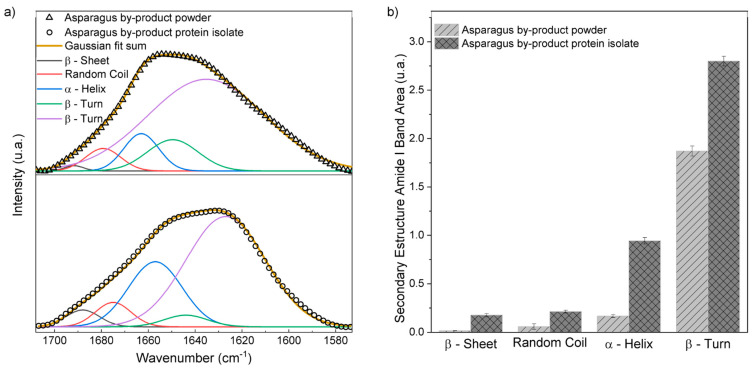
(**a**) Gaussian adjustments of the amide I band of the powder and protein isolate from asparagus by-products; (**b**) area of the bands resulting from the Gaussian adjustment of the amide I band of the powder and protein isolate from asparagus by-products (*n* = 3). Different uppercase letters are significantly different (*p* < 0.05) for the asparagus by-product powder. Different lowercase letters are significantly different (*p* < 0.05) for the asparagus by-product protein isolate.

**Table 1 foods-13-00894-t001:** Centesimal composition and soluble protein fractions of asparagus by-product powder.

Parameters	(%)
Moisture	8.56 ± 0.05
Ash	5.84 ± 0.08
Lipids	2.23 ± 0.19
Crude fiber	33.17 ± 0.39
Protein	15.65 ± 0.26
Carbohydrates	34.54 ± 0.53
Fractions	Protein (%)
Albumin	14.58 ± 0.52 ^c^
Globulin	11.96 ± 0.28 ^d^
Prolamin	14.59 ± 0.36 ^c^
Glutelin	22.88 ± 0.13 ^b^
Insoluble residues	35.98 ± 0.01 ^a^

Values are mean ± standard deviation of three replicates. Different letters in the same column are significantly different (*p* < 0.05).

**Table 2 foods-13-00894-t002:** Box–Behnken design for pH, time, and temperature and their response, content protein (mg protein.g^−1^ of asparagus by-product).

Independent Parameters (Coded Value)				Protein Content, Y (mg·g^−1^)
Run	pH (X_1_)	Time (X_2_, min)	Temperature (X_3_, °C)	Experimental Results
1	8 (1)	20 (−1)	50 (0)	110.57
2	10 (1)	20 (−1)	50 (0)	116.30
3	8 (−1)	60 (1)	50 (0)	111.70
4	10 (1)	60 (1)	50 (0)	122.76
5	8 (−1)	40 (0)	30 (−1)	112.59
6	10 (1)	40 (0)	30 (−1)	120.42
7	8 (−1)	40 (0)	70 (1)	114.61
8	10 (1)	40 (0)	70 (1)	122.92
9	9 (0)	20 (−1)	30 (−1)	116.55
10	9 (0)	60 (1)	30 (−1)	117.76
11	9 (0)	20 (−1)	70 (1)	121.87
12	9 (0)	60 (1)	70 (1)	119.45
13	9 (0)	40 (0)	50 (0)	130.99
14	9 (0)	40 (0)	50 (0)	130.19
15	9 (0)	40 (0)	50 (0)	132.45
16	9 (0)	40 (0)	50 (0)	133.01
17	9 (0)	40 (0)	50 (0)	130.83

**Table 3 foods-13-00894-t003:** Estimated regression coefficients for a quadratic model and ANOVA for experimental results from asparagus by-product protein optimization.

Source	Sum of Squares	Degree of Freedom	MEAN SQUARE	*F* Value	*p* Value
X_1_	123.59	1	123.59	88.28	0.0007 *
X_2_	9.75	1	9.75	6.96	0.0577
X_3_	12.90	1	12.90	9.21	0.0386 *
X_1_X_2_	7.09	1	7.09	5.07	0.0875
X_1_$X_{2}^{2}$	0.05	1	0.05	0.04	0.8564
$X_{1}^{2}$X_2_	9.67	1	9.67	6.91	0.0583
X_1_X_3_	0.06	1	0.06	0.04	0.8478
$X_{1}^{2}$X_3_	0.78	1	0.78	0.56	0.4962
X_2_X_3_	3.30	1	3.30	2.36	0.1996
$X_{1}^{2}$	319.80	1	319.80	228.43	0.0001 *
$X_{2}^{2}$	233.31	1	233.31	166.65	0.0002 *
$X_{3}^{2}$	111.40	1	111.40	79.57	0.0009 *
Pure Error	5.6	4	1.4		
Total SS	921.8923	16			
R-squared				0.994	
Adj R-squared				0.976	

X_1_, X_2_, and X_3_ are the dimensionless coded values of pH, time, and extraction temperature, respectively. * Terms with significant effect (*p* < 0.05).

**Table 4 foods-13-00894-t004:** Extraction and color parameters, and technofunctional, bioactive, and antioxidant properties of the asparagus by-product protein isolate.

Parameters	
Protein content (%)	90.48 ± 1.94
Extraction efficiency (%)	84.44 ± 0.67
Protein yield (%)	43.47 ± 1.23
Color parameters	
L*	31.91 ± 0.36
a*	−0.01 ± 0.08
b*	−2.11 ± 0.34
Technofunctional Properties	
Water absorption capacity (g·g^−1^)	4.49 ± 0.39
Oil absorption capacity (g·g^−1^)	3.47 ± 0.11
Bioactive and antioxidant properties	
Total polyphenol content (TPC) (mg GAE·g^−1^)	7.29 ± 0.58
ABTS assay (TEAC μmol.100g^−1^)	145.76 ± 0.76
DPPH assay (TEAC μmol.100g^−1^)	65.21 ± 0.38

Values are means ± SD of triplicate determinations.

## Data Availability

The original contributions presented in the study are included in the article, further inquiries can be directed to the corresponding author.

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
