# Peer review of "Optimization and Characterization of Protein Extraction from Asparagus Leafy By-Products"

_foods, 2024, doi:10.3390/foods13060894_

Round 1

Reviewer 1 Report

Comments and Suggestions for Authors

Dear Authors,

The current paper entitled Optimization and characterization of protein extracted from asparagus leafy by-products, is interesting and nicely written.

Some minor observations/suggestion for the paper, can be found below:

In the introduction section, lines 45-46, please eighter use FAO as a reference, or place a different one for that statement.

In the introduction section, general potential usage of asparagus is missing. If previous applications were done in animals feeding, it seems that the by-products have a general utilization. Maybe if the study will have a hypothesis, will make clearer the background and the purpose.

Line 82, dried how? Oven? Sun? please describe in more detail the drying process.  

Line 89, same observation for incineration, provide the name of the ash oven and for how long.

In the entire paper please use by-products instead of byproducts.

Table 3, left column, the variants are not visible, due to the page numbering, please correct.

The amino acids determined in the asparagus by-products, should be presented as values in table, and the chromatogram separately, the amino acids which were not determined should be mentioned as other AA. Also, they should have a measurement unit. Please correct.

Figure 5 A and B, are presented quite wrong. Figure 5B is not well visible, or the information presented there. It should be presented side by side with figure 5A.

Same observation for figure 6B. It is not visible not the information presented.

Author Response

Reviewer: 1

Comments and Suggestions for Authors:

- In the introduction section, lines 45-46, please eighter use FAO as a reference, or place a different one for that statement.

A: Motivated by your comment, we added the reference (page 2, lines 46-47).

- In the introduction section, general potential usage of asparagus is missing. If previous applications were done in animals feeding, it seems that the by-products have a general utilization. Maybe if the study will have a hypothesis, will make clearer the background and the purpose.

A: Thank you for your contribution, we have added the general potential use of asparagus and a hypothesis to the paper (page 2, lines 73-79).

- Line 82, dried how? Oven? Sun? please describe in more detail the drying process.  

A: In line 87-88 (page 2) the drying process was described in more detail.

- Line 89, same observation for incineration, provide the name of the ash oven and for how long.

A: In line 96 (page 2) the name of the oven was informed, as well as the time the samples remained until complete incineration.

- In the entire paper please use by-products instead of byproducts.

A: The term by-products was standardized throughout the paper.

- Table 3, left column, the variants are not visible, due to the page numbering, please correct.

A: Table 3 was formatted with the aim of making all variants visible.

- The amino acids determined in the asparagus by-products, should be presented as values in table, and the chromatogram separately, the amino acids which were not determined should be mentioned as other AA. Also, they should have a measurement unit. Please correct.

A: Thank you sincerely for your comment, but the amino acid analysis was not carried out in a quantitative way, more comparatively with a standard with known amino acids, therefore the amino acids were not tabulated because the amount of each amino acid was not quantified.

- Figure 5 A and B, are presented quite wrong. Figure 5B is not well visible, or the information presented there. It should be presented side by side with figure 5A.

A: Figure 5A and 5B were placed side by side for better visualization.

- Same observation for figure 6B. It is not visible not the information presented.

A: To improve the visualization of the information presented, figures 6A and 6B were placed side by side.

Reviewer 2 Report

Comments and Suggestions for Authors

I have some comments to revise the manuscript. Please see the attachment.

Comments on the Quality of English Language

English language has to be modified significantly. Many words are combined together, which makes it difficult to understand. The sentences needs to be concised.

Author Response

Reviewer: 2

Comments and Suggestions for Authors:

- Line 15-17: In recent decades, it has been observed that by-products……I think this sentence is not needed.

A: Motivated by your comment, we deleted the sentence.

- Line 21-25: Please paraphrase the sentence. Its too big to understand.

A: The sentence was paraphrased (page 2, lines 22-27).

- Line 46: the global demand for food could by 50%, please correct the sentence.

A: The sentence was corrected (page 2, lines 47).

- Line 54: protein nutritional needs [4]……

A: Thanks. We apologize for the typing error (page 2, line 56).

- Line 76: Use RSM as an abbreviation

A: It was corrected (page 2, line 80).

- Line 93: The protein fractionation of the asparagus byproduct was based on its solubility in 93 water, saline, alkaline and hydroalcoholic solutions, reference??

A: The reference was at the end of the paragraph; we placed it in the suggested location (page 3, line 101).

- Line 112: conditionsand???

A: We apologize for the typing error, it has been corrected (page 3, line 119).

- Line 220: Provide the reference.

A: We provide the reference (page 5, line 237).

- Line 229: analyzed according to [18]……this reference name has to be mentioned.

A: The reference name was mentioned (page 5, line 243). 

Line 262: behavior was observed?????

A: The sentence was rewritten, the behavior referred to is that of the protein fractions extracted from radish leaves (Raphanussativus L.), where glutelins were superior to the others, as well as in asparagus by-product powder (page 6, lines 280-282).

- Line 387: we have the mass spectrum obtained in which the…..this sentence does not seems scientific.

A: The sentence was rewritten (page 11, line 487).

- Line 460: Ramdom coil?

A: We apologize for the typing error (page 14, line 470).

- Line 477-479: I think this is not required.

A: Motivated by your comment, we identified that the sentence “The pH has an important effect on the pigments (as chlorophyll, carotenoids, anthocyanins, myoglobin, etc.) responsible for the color of fruits, vegetables and meat. Chlorophylls, pigments responsible for the characteristic green color of asparagus [37]” is really not necessary. 

- Line 490: were shown in Table 4…..

A: It was corrected (page 14, line 498).

- Line 513: The optimal processing conditions using the optimization method were obtained when the extraction pH was 9, time was 40 min and temperature was 50 °C, please revise the sentence.

A: Motivated by your comment, we have reviewed and rephrased the sentence (page 15, lines 521-522).

- Line 516: protein extraction method for asparagus by-product………

A: Thanks for the comment, we replaced the term "from" with the term "for" (page 15, line 525).

Reviewer 3 Report

Comments and Suggestions for Authors

This study used response surface methodology to optimize the process of extracting proteins from asparagus leafy by-products and evaluated the protein composition and function. It is a very interesting paper due to the determination of amino acid composition using mass spectrum, and analysis of protein structure.

1)     Line 21, the isolate obtained presented 90.48% protein with 43.47% protein yield. Why not to decide amino acid composition in the isolate using an amino acid analyzer or HPLC method? Only six amino acids were determined in this study, these species is much below the common species of 17 in plant extracts.

2)     Table 3 is not clear.

3)     In figure 3, the protein isolate from asparagus by-product is an alkaline protein. On lines 108-113, the protein isolate was extracted by water with pH 8-10, the extraction solution should be given in the abstract and conclusion sections.

4)     In the abstract, ABTS and TEAC should be noted, not in abbreviations.

5)     This paper is useful for giving a process that extracts alkaline proteins from asparagus by-product and then precipitates them using isoelectric point of pH 3.2. The economic feasibility should be given in the discussion and conclusion sections.

Author Response

Reviewer: 3

Comments and Suggestions for Authors:

 - Line 21, the isolate obtained presented 90.48% protein with 43.47% protein yield. Why not to decide amino acid composition in the isolate using an amino acid analyzer or HPLC method? Only six amino acids were determined in this study, these species is much below the common species of 17 in plant extracts.

A: In the qualitative amino acid analysis methodology used in this paper, no type of fragmentation or alteration was made, the sample was analyzed integrally, that is, only the free amino acids in the sample were identified. Amino acids, peptides and proteins may be present in the sample, but in the methodology used they were not identified. In the future, the composition of amino acids will be analyzed.

- Table 3 is not clear.

A: Table 3 has been formatted.

- In figure 3, the protein isolate from asparagus by-product is an alkaline protein. On lines 108-113, the protein isolate was extracted by water with pH 8-10, the extraction solution should be given in the abstract and conclusion sections.

A: Thank you for your consideration, the protein from asparagus by-products was extracted in an aqueous solution, and this information has been added in the abstract and conclusion sections [page 1 (line 19) and page 15 (line 520)].

- In the abstract, ABTS and TEAC should be noted, not in abbreviations.

A: ABTS and TEAC were noted without abbreviation in the abstract (page 1, lines 24-25).

- This paper is useful for giving a process that extracts alkaline proteins from asparagus by-product and then precipitates them using isoelectric point of pH 3.2. The economic feasibility should be given in the discussion and conclusion sections.

A: The economic feasibility study on the use of asparagus leafy by-products as sources of protein for human consumption was not carried out in this work, but may be addressed in future studies. This information was added in the conclusion section (page 15, lines 527-530).
